# How to Improve Non-Invasive Diagnosis of Endometriosis with Advanced Statistical Methods

**DOI:** 10.3390/medicina59030499

**Published:** 2023-03-03

**Authors:** Maria Szubert, Aleksander Rycerz, Jacek R. Wilczyński

**Affiliations:** 1Department of Surgical and Oncological Gynecology, 1st Department of Gynecology and Obstetrics, Medical University of Lodz, M. Pirogow’s Teaching Hospital, Wilenska 37 St., 94-029 Lodz, Poland; 2Club 35, Polish Society of Gynecologists and Obstetricians, ul. Cybernetyki 7F/87, 02-677 Warszawa, Poland; 3Faculty of Mathematics and Computer Science, University of Lodz, Banacha 22, 90-238 Lodz, Poland

**Keywords:** endometriosis, non-invasive diagnosis, machine learning algorithm, CA125, LASSO

## Abstract

*Background and Objectives*: Endometriosis is one of the most common gynecological disorders in women of reproductive age. Causing pelvic pain and infertility, it is considered one of the most serious health problems, being responsible for work absences or productivity loss. Its diagnosis is often delayed because of the need for an invasive laparoscopic approach. Despite years of studies, no single marker for endometriosis has been discovered. The aim of this research was to find an algorithm based on symptoms and laboratory tests that could diagnose endometriosis in a non-invasive way. *Materials and Methods*: The research group consisted of 101 women hospitalized for diagnostic laparoscopy, among which 71 had confirmed endometriosis. Data on reproductive history were collected in detail. CA125 (cancer antigen-125) level and VEGF1(vascular endothelial growth factor 1) were tested in blood samples. Among the used statistical methods, the LASSO regression—a new important statistical tool eliminating the least useful features—was the only method to have significant results. *Results*: Out of 19 features based on results of LASSO, 7 variables were chosen: body mass index, age of menarche, cycle length, painful periods, information about using contraception, CA125, and VEGF1. After multivariate logistic regression with a backward strategy, the three most significant features were evaluated. The strongest impact on endometriosis prediction had information about painful periods, CA125 over 15 u/mL, and the lowest BMI, with a sensitivity of 0.8800 and a specificity of 0.8000, respectively. *Conclusions*: Advanced statistical methods are crucial when creating non-invasive tests for endometriosis. An algorithm based on three easy features, including painful menses, BMI level, and CA125 concentration could have an important place in the non-invasive diagnosis of endometriosis. If confirmed in a prospective study, implementing such an algorithm in populations with a high risk of endometriosis will allow us to cover patients suspected of endometriosis with proper treatment.

## 1. Introduction

Endometriosis is one of the most common diseases in women who suffer from fertility problems or pelvic pain. The prevalence of endometriosis is really unknown. The disease is found in about 0.5–5% of women of reproductive age, mostly in those (25–40%) with fertility problems. Some authors reported a more frequent occurrence of the disease. The most difficult form to diagnose without invasive methods is peritoneal endometriosis. Other forms such as endometrioma or deep endometriosis are mostly recognizable through imaging methods [1,2]. The ovarian form of endometriosis, so-called endometrioma, can be easily diagnosed by ultrasound and there are also certain findings of deep endometriosis in ultrasound or in magnetic resonance. Peritoneal endometriosis includes highly visible dark scarred, as well as more difficult-to-recognize nonpigmented, subtle forms. Associated histological features present in some cases include reactive mesothelial proliferation, inflammation, haemosiderin pigment deposition or microcalcification, and decidualization. These changes are hormonal-dependent [3]. The presence of peritoneal lesions and their variability during the menstrual cycle can cause pelvic pain, recurring in the secretory phase of the menstrual cycle, pain during menses, profuse menses, painful intercourses, dyschezia and several other symptoms that are sometimes initially diagnosed as other organs diseases such as irritable bowel disease or recurrent urinary tract infections [4,5]. The severity of symptoms is a common cause of work absence. This fact and the necessity for invasive procedures affect social costs associated with endometriosis.

Despite many studied markers and proteins in serum, plasma, peritoneal fluid, urine, and endometrium biopsies, any of the known particules can be used in the diagnosis of peritoneal endometriosis alone [6,7,8]. Currently, researchers strive to find groups of factors that, in particular combinations, would provide a tool to diagnose endometriosis, especially its early stages, with maximum sensitivity and specificity in a non-invasive way [9,10]. 

Disturbed local immunological and hormonal response as well as angiogenesis are dominant factors in the development of endometrium foci in the peritoneal cavity. Stem cell markers, disturbed gene expression, and dysregulated iron homeostasis are also taken into consideration not only in the experimental studies on the pathophysiology of endometriosis but also in many other aspects of diagnosis and treatment of endometriosis, as well as in deliberations about its potential role in carcinogenesis [11,12,13,14,15]. Many researchers attempted to use inflammatory or vascular markers to develop a non-invasive test for endometriosis. Guralp O. et al. have recently published a test combined with CA125, endocan, YKL40, and copeptin, that diagnosed endometriosis with over 90% sensitivity [16]. Dorien F. et al. [17] conducted a study to validate their own model for a non-invasive test published previously [18]. The tested biomarkers model consisted of CA125, VEGF, annexin V, and glycodelin/sICAM-1 on a new group of patients, and they were not able to confirm previously reported high sensitivity (82%) and specificity (75%) in peritoneal endometriosis. However, they concluded that CA125 remained the most important protein to increase sensitivity and specificity of non-invasive diagnostic models, with the mean value for controls set at 15 U/mL [17]. These suggestions were also confirmed in reviews, systematic reviews, or meta-analyses [19,20]. CA125 is a glycoprotein also known as Mucin-16 (the largest membrane-bound mucin) encoded by the *MUC16* gene [21]. Because the rise in CA125 is unspecific, it cannot serve as a biomarker for endometriosis alone, but its role in scientific research is well documented; however, the cut-off value of CA125 differs from 15 U/mL [17], through 20 U/mL to 85 U/mL according to several multi-center studies and meta-analyses [22,23,24]. 

Researchers also tried to use newly developed statistical tools to achieve more sensitivity and specificity in the diagnosis of endometriosis. Data mining techniques, first described and widely used in business or geophysics, generate descriptions and predictions about a target dataset. They can either describe this dataset, using a lot of features, or they can predict outcomes through the use of machine learning algorithms. As Curchoe CL. et al. stated, the era of artificial intelligence, which also encompasses machine learning, has already begun and is developing very fast, especially in cancer and fertility studies [25]. It is well known that properly-used new statistical methods could shed new light on the old data. 

To date, there are only several authors that applied a data mining approach in endometriosis. In 2013, Wang Y. et al. and Cheng M. et al. published the first data mining elaboration on recurrent ovarian endometrial cysts and the possibility of curing them with ethanol sclerotherapy [26,27]. 

Akter S. et al. analyzed 38 RNA-seq transcriptomics samples (16 endometriosis and 22 controls) and identified five genomic signatures as potential biomarkers; then, using a decision tree algorithm constructed a model with higher predictive performance [28]. Bendifallah S. et al. recently proved that artificial intelligence and machine learning algorithms based on endometriosis symptoms can be easily used as screening tools for general practitioners or even patients themselves [29]. In our previous study, we tried to find a non-invasive algorithm to diagnose endometriosis based on several biomarkers from endometrium but we reached only 60% sensitivity [30]. 

Hence, the new statistical methods are now available in medicine, we tested the gathered data once again, with several advanced statistical models among which Lasso regression had the highest sensitivity. The aim of the current study was to find an easy algorithm that could help to diagnose peritoneal endometriosis without surgical interventions.

## 2. Methods

The study was a retrospective analysis of data with the use of the new statistical tools-data gathered for the purpose of prospective research on angiogenesis in endometriosis, the detailed results of which were published elsewhere [30]. The study titled “Angiogenesis and Inflammatory Response in Endometriosis before and after Danazol Treatment” was published in 2014. The development of new statistical methods motivated us to recalculate the sensitivity and specificity of the gathered data to find if their application can improve diagnostic accuracy. Informed consent for using anonymized data was given by the patients also for further analyses (RNN/11/09/KE). Bioethics committee opinion was obtained for the use of retrospective data and the principles of the Declaration of Helsinki were followed while carrying out the present study.

### 2.1. Population

The research group consisted of 101 women hospitalized in 2010 for diagnostic or diagnostic and therapeutic laparoscopy for pelvic pain, or pelvic pain with coexisting infertility or suspicion of endometriosis due to other reasons. Exclusion criteria were: previous hormonal treatment, chemotherapy, or biologic treatment in the 3 months prior to the laparoscopy, any coexisting cancer or other severe diseases (hypertension, diabetes, autoimmunologic diseases, immune deficiency diseases), any kind of inflammation during or 1 month prior to the laparoscopy (e.g., respiratory tract infections, urinary or genital tract infections), imaging findings of large myomas, uterine abnormalities or other forms of endometriosis (deep endometriosis, cesarean scar endometriosis, endometrioma). Infertility alone was not an indication for laparoscopy in the studied population. In 71 patients, endometriosis was diagnosed. Only patients with peritoneal endometriosis were recruited, with mild endometriosis (stage I and II according to the rASRM scale) in 53 patients and severe endometriosis (stage III) in 18 patients. To assess endometriosis, the rASRM scale (revised American Society of Reproductive Medicine scale) was implemented. The rASRM scale is probably the best-known classification around the world for peritoneal endometriosis. Numerical values are assigned to endometriosis lesions according to their location, size, and presence of adhesions. Adhesions are also subjected to the scoring with the highest number of points for the dense adhesions that obliterate the posterior cul-de-sac. Finally, all of the points assigned are summed, and the resulting point scores are classified into four grades of severity. In our study, none of the patients presented with the IV stage according to rASRM. 

The remaining 30 patients subjected to laparoscopy had negative confirmation of endometriosis and were diagnosed with benign lesions such as small dermoid cysts, small peritubal cysts, peritoneal adhesions, and small myomas or were simply without any visible pathologies. During admission to the hospital, each patient fulfilled the questionnaire about possible symptoms combined with the menstrual cycle. Vital signs and BMI (body mass index) were collected. We collected also a history of onset of menstrual bleeding, menstrual pattern, length of the cycle and length of the bleeding, pain during the secretory phase of the cycle, painful intercourses, and menses. The pain was assessed according to the VAS scale, for each kind of pain separately: menstrual pain, pain during cycle, and pain during intercourses. The interview also asked about a woman’s reproductive history, other diseases or surgeries, and contraception For the angiogenesis study, a blood sample was collected to determine the concentration of several angiogenesis markers and CA125 level. The inclusion into the study or control group followed the laparoscopy findings. Because in the previous study only CA125 and VEGF-1 had the potential to differentiate the study and control group, we chose these two markers for our statistical analysis to check their usefulness in advanced statistical methods. 

### 2.2. Statistical Analysis

Data were tested in several statistical methods approved in medicine in recent decades: decision trees, naive Bayes, the K-means method, and LASSO regression. Among them, LASSO was the most efficient method, with the highest sensitivity and specificity. Decision trees, naive Bayes, and the K-means method had low areas under the curve (AUC), positive predictive values (PPV), negative predictive values (NPV), and sensitivity. LASSO regression is a kind of regression that uses a technique “shrinkage” where the coefficients of determination are shrunk towards zero. In LASSO regression, less important features of the dataset are eliminated one by one. Thus, it provides us with the benefit of feature selection and simple model creation. The RStudio 4.2.0 program (RStudio, Boston, MA, USA)was used for the statistical analysis. The threshold for statistical significance was set to *p* < 0.05. In the first step, the model finds interdependence between variables in the so-called “training” group. Patients were randomly assigned to the training group (*n* = 71) and the testing group (*n* = 30) (Table 1.) Variables indicated in LASSO were further examined by backward stepwise multivariate logistic regression to find the best diagnostic model. Based on the obtained diagnostic model, we made predictions for both groups, and then we calculated the accuracy, sensitivity, specificity, positive predictive value (PPV), and negative predictive value (NPV).

## 3. Results

The mean age of the patients was 31.99 years (SD 6.09; min. 18; max. 47). Painful periods (VAS ≥ 4) were reported by 53 women with endometriosis (76.81% of the study group) and 14 without endometriosis (43.75% of the control group). The t-student test showed a statistically significant difference in CA125 level between women with and without endometriosis (Table 2 and Figure 1).

First, we performed LASSO regression methods with cross-validation on all groups to minimalize the number of variables. Out of 19 features based on the results of LASSO, we chose seven variables: body mass index, age of menarche, cycle length, painful periods, information about using contraception, CA125, and VEGF1 (Figure 2). Using multivariate logistic regression with backward strategy, we were able to reduce features to the three most significant. The diagnostic model was based on: BMI, painful periods, and CA125 level. The strongest impact on endometriosis prediction had information about pain during periods and the lowest BMI. Based on the testing group, we calculated the sensitivity and specificity as 0.8800 and 0.8000, respectively. The area under the ROC curve (receiver operating characteristic curve) was 0.8400 and the positive and negative predictive values were 0.9565 and 0.5714. 

We plotted the ROC curve for parameters included in the diagnostic model. The area under the ROC curve for painful periods was 0.7220 and for the CA125 level 0.8000 (Figure 3). We set the CA125 level above 15 U/mL as a differentiating between the study and control groups. The sensitivity and specificity for CA125 were 0.6000 and 1.0000, respectively. The positive predictive value was the lowest of testing parameters and model–0.9130 (Figure 4).

## 4. Discussion

We developed the algorithm to diagnose endometriosis in a non-invasive way and tested it in several statistical methods among which LASSO regression was the most efficient one. Only painful periods, BMI level, and CA125 concentration influenced the diagnosis of endometriosis. At first, LASSO was described in geophysics and almost twenty years later in medicine [31]. Introducing LASSO in the statistics was in order to improve the prediction accuracy and interpretability of regression models. LASSO was proved to predict accuracy better compared to the widely used stepwise regression model. A broadly used stepwise regression model could increase prediction error by choosing only variables with strong relationships with the outcome. Other new statistical tools are less prone to prediction errors, among them are decision trees and neural networks as well as machine learning algorithms that can use several statistical methods such as the naive Bayes method. Decision trees (DT) are popular machine learning models applied to both classification and regression tasks; however, sometimes they can be like real trees—too many branches cause “noise” in the data. Bayesian statistics is thought to provide a probabilistic framework that reduces the instability of the data. It can also omit limitations of traditional statistics when comparing parametric and non-parametric data, especially in endometriosis in which symptoms, the clinical experience of the doctor, and the patient’s needs are the most important factors in the diagnosis and treatment [32]. Looking for a non-invasive test for endometriosis, one should nowadays keep these new statistical tools in mind. 

To date, Eskenazi B. et al. proved that only ovarian endometriosis, but not nonovarian endometriosis, could be reliably predicted with noninvasive tools [33]. Ultrasound and examination best predicted ovarian endometriosis, correctly classifying 100% of cases with no false positive diagnoses in the study sample. Similar results were found in the test sample. The decision tree given by them, unfortunately, predicted nonovarian endometriosis only very poor. What is more, even enhancing the decision tree model by adding symptoms to ultrasound examination, enabled us to correctly diagnose the affected patients in only 38%. Our results are in accordance with Bendifallah et al. [29], who demonstrated that machine learning algorithms (MLAs) based on 16 clinical and symptom-based features enabled diagnosis and early prediction of endometriosis onset. Despite the survey nature of their study and missing data in the questionnaires, their validation of the MLA technique questions laparoscopy as a basic screening tool for endometriosis. 

There have already been several attempts to introduce questionnaires into general practice to fasten the diagnosis of endometriosis but many of them were not properly validated or were only approved to use as PRO (patient-reported outcomes) in clinical studies. Gater A. et al. summarized the Endometriosis Symptom Diary (ESD) and the Endometriosis Impact Scale (EIS) and supported the content validity [34] but they were lacking advanced statistical methods. These methods could be very useful when combining symptoms with endometriosis markers. After the Cochrane review [35] in 2016, there was a noticeable reverse from studies on markers in endometriosis, but the last few years have been promising in this field. Despite the negative findings of the mentioned Cochrane review [35], their authors proposed the sensitivity and specificity cut-offs of possible tests to rule-in or rule-out the diagnosis of endometriosis. Although no panel currently meets the Cochrane rule-in criteria, CA125 is still near the value set [36]. According to a meta-analysis by Hirsch M. et al., its sensitivity is 52% and its specificity is 93% [37]. Researchers are now assured that one single marker of endometriosis does not exist and CA125 could only serve as a potential marker in statistical deliberations. 

New studied targets as endometriosis markers are now miRNAs (small non-coding RNAs that regulate many processes) as well as stem cells [38,39]. The data indicate that there is probably so-called “miRNA endometriosis signature”—over 1000 miRNAs that are up or down regulated, typically for endometriosis. Bendifallah S. et al. conducted the first prospective study to report a saliva-based diagnostic miRNA signature for endometriosis [40]. To assess so huge an amount of data and to combine this data with clinical symptoms in one algorithm, new statistical tools are needed. As already mentioned, it is possible nowadays to use advanced statistical methods in medicine: decision trees, neuronal networks, and artificial intelligence. 

Our study, with a very easy dataset, confirms that advanced statistical methods are crucial while creating non-invasive tests for endometriosis. In our study, the proposed algorithm reached satisfactory sensitivity with only three easy features: painful menses, BMI level, and CA125 concentration. This needs to be confirmed in other populations and prospective studies. A strong point of the study was the laparoscopic confirmation of endometriosis and exclusion of the peritoneal disease and other diseases during the preoperative ultrasound scan and surgical procedure. Questions during anamnesis were easy and understandable. However, there are crucial limitations, and one should interpret the results carefully. The major limitations include the population with a 71% prevalence of endometriosis; the exclusion of inflammatory diseases, cancer, myomata, or other sources of positive CA125, and the retrospective choice of a statistical method. The algorithm cannot be extended to the adolescent population nor to women after hormonal treatment; women with cancer or after cancer treatment; large myomata; uterine abnormalities; deep endometriosis or endometriomas on imaging or women suspected of pelvic inflammatory disease. These populations were not recruited into our study. These conditions, however, although they may cause pain in the pelvis, are rather easy to diagnose after proper differential diagnostic procedures. The infertile population without other symptoms is also not appropriate to test with this algorithm because of the priority to conceive and not to treat pain. A high percentage of endometriosis patients in our population is probably influenced by the specificity of care provided by the clinic. One should bear in mind that the proposition of an algorithm to diagnose endometriosis is rather to help to select affected women to schedule them for first-line treatment without any delay. In our opinion, such an algorithm could be an objective tool, the results of which the doctor can show to the patient to convince her of hormonal treatment. Broadly known facts are that some women are afraid of hormonal treatment due to its side effects [41]. Horne AW. et al. comprehensively discussed selection bias as a bias that is present in almost all one-center studies and that is influenced by many economical and geographical factors [42]. Survey studies are also prone to selection bias, but our whole population was surgically tested only after the survey had been performed. There are other studies on larger populations where the control group was not scheduled for laparoscopy and patients were only recruited because of lack of endometriosis symptoms [43,44]. Hence, there is a certain percentage of women with endometriosis but without pain, the methodology of the endometriosis studies is still the hottest question when seeking non-invasive markers of the disease [45]. 

Preparing an easily accessible online algorithm will be the next promising step in the diagnosis of endometriosis that could hasten the diagnosis and treatment. One of the most recent trends in endometriosis is to hormonally block the pituitary–ovarian axis to alleviate symptoms combined with the menstrual cycle, instead of scheduling patients for invasive laparoscopy. A reduction in symptoms could act as proof of the presence of endometriosis [46]. This approach, however, is not accepted by all patients and cannot rule out endometriosis for sure, especially in adolescents [47]. Typically, patients should be offered combined hormonal contraceptives (oral contraception = OC) or progestins. Analogs of GnRH are also proven to alleviate endometriosis pain [48]. Contraindications should be taken into consideration as well as local regulations regarding monthly costs paid by women for the therapy. Possible side effects should also be discussed, especially with patients who did not use OC previously or are scheduled for GnRH agonists or antagonists therapy. As there are several meta-analyses that concluded that using OC results in a statistically significant reduction in endometriosis-related pain, resulting in improvement in quality of life [49,50], the hormonal treatment should be offered as first-line therapy, even instead of laparoscopy, which is invasive and cost-consuming. On the other hand, visualization of the disease and its superficial peritoneal lesions could have a positive psychological impact on the patient’s attitude to the disease [49]. There are controversies now about what to offer if the first-line treatment fails. GnRh (gonadoliberin hormone) agonists or antagonists which are broadly used as second-line treatment, could have severe side effects. At this point, laparoscopy is probably a better option than empirical treatment, even supported by the algorithm created for non-invasive diagnostics. Hence, the endometriosis patient is usually very aware of the disease but also suffers from chronic stress [51], and all pros and cons regarding surgical diagnosis and life-long treatment of endometriosis should be carefully discussed. 

CA125 had in, in our study, the highest potential to determine the risk of endometriosis; therefore, it could be useful while creating a non-invasive algorithm to diagnose endometriosis. Although there are studies that do not confirm the role of CA125 in endometriosis [46], one had to remember that the CA125 level that could determine the risk is far below the upper range for other diseases [36]. If we ever find a biomarker or set of markers for endometriosis, testing a patient with dysmenorrhea, pelvic pain, or other typical symptoms of endometriosis with the non-invasive algorithm will always be the cheapest and most easily available option compared to laparoscopy. 

## 5. Conclusions

Advanced statistical methods are crucial when creating non-invasive tests for endometriosis. The current use of Lasso analysis identified the combination of painful menses, BMI levels, and CA125 as predictive of endometriosis with sensitivity and specificity of 0.88 and 0.80, respectively, in a retrospective study of a population with a 70% prevalence of endometriosis and other cause of CA125 elevation excluded.

An algorithm based on three easy features of painful menses, BMI level, and CA125 concentration with the LASSO statistical tool could have an important place in the diagnosis of endometriosis, especially in outpatient settings. Hence, since patients with endometriosis symptoms should be scheduled for diagnostic laparoscopy only if hormonal treatment does not alleviate symptoms [46], the positive results of the aforementioned algorithm can facilitate the decision to implement hormonal treatment. 

## Figures and Tables

**Figure 1 medicina-59-00499-f001:**
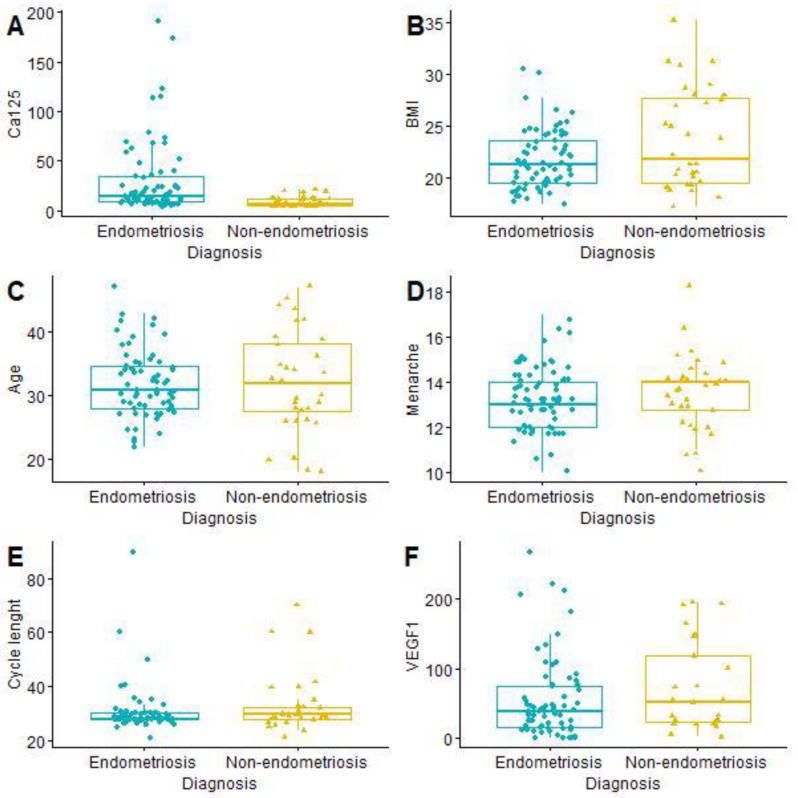
Comparison of continuous features from LASSO and age: CA125 (**A**); BMI (**B**); Age (**C**); Menarche (**D**); Cycle length (**E**); VEGF1 (**F**). BMI—body mass index, CA125—cancer antigen 125, VEGF1—vascular endothelial growth factor 1.

**Figure 2 medicina-59-00499-f002:**
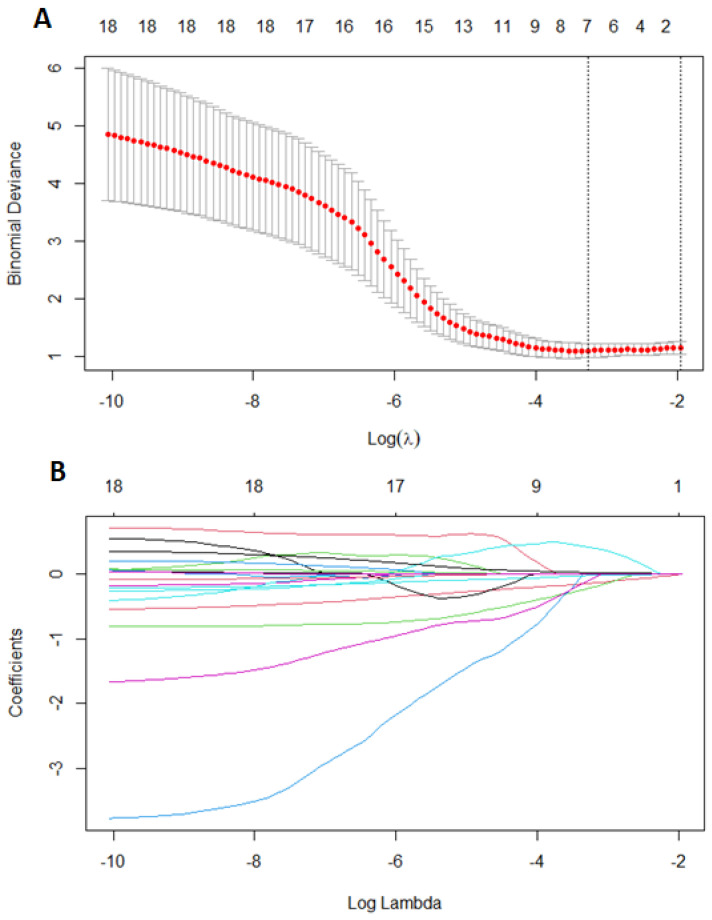
Features selection using least absolute shrinkage and selection operator regression (LASSO). Ten-fold cross-validation for tuning parameter selection in the LASSO (**A**), coefficients of 18 features in LASSO (**B**). Each line represent one feature.

**Figure 3 medicina-59-00499-f003:**
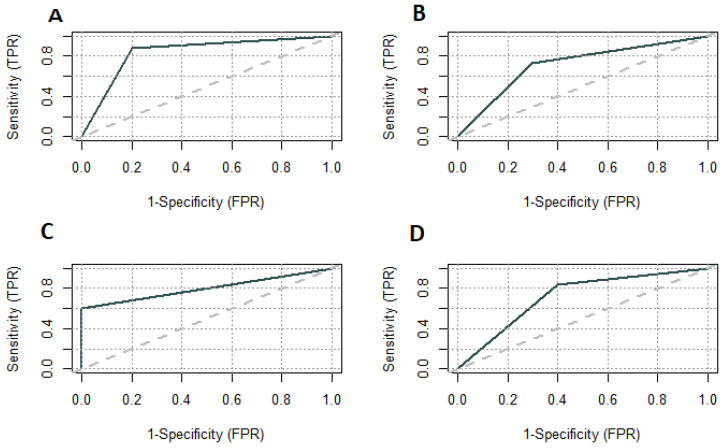
ROC curves for model on testing group (**A**); model on training group (**B**); CA125 on testing group (**C**); Painful periods on testing group (**D**). TPR—true positive rate; FPR—false positive rate.

**Figure 4 medicina-59-00499-f004:**
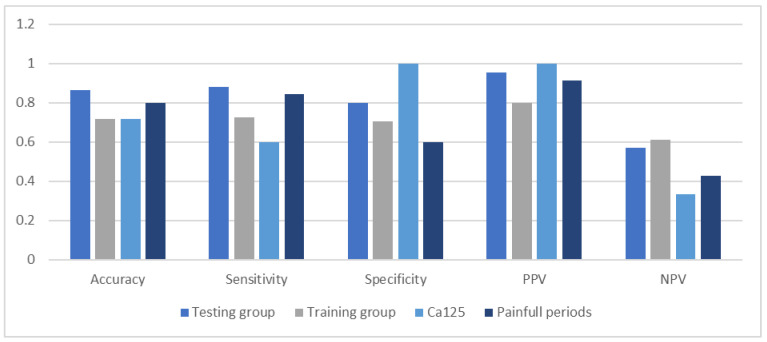
Comparison of accuracy, sensitivity, specificity, and predictive values of the diagnostic model.

**Table 1 medicina-59-00499-t001:** The number of samples randomly assigned to training (70%) and testing (30%) group. In the “training” group the test finds interdependence between variables which is later checked in the “testing” group.

Class	Training Group (*n* = 71)	Testing Group (*n* = 30)
Endometriosis	44	25
Non-endometriosis	27	5

**Table 2 medicina-59-00499-t002:** Quantitative description of continuous features from LASSO regression and age. The seventh feature–painful periods was non-continuous and therefore is not shown below. BMI—body mass index, CA125—cancer antigen 125, VEGF1—vascular endothelial growth factor 1.

	All Patients (Mean)	Endometriosis	Non-Endometriosis	*p* Value
Age (years)	31.99 (SD = 6.09)	31.87 (SD = 5.10)	32.25 (SD = 7.93)	0.8056
Menarche (year)	13.40 (SD = 1.41)	13.35 (SD = 1.33)	13.53 (SD = 1.59)	0.5694
BMI (kg/m^2^)	22.39 (SD = 3.65)	21.79 (SD = 2.82)	23.70 (SD = 4.80)	0.0422
CA125 (U/mL)	23.47 (SD = 32.79)	29.97 (SD = 37.64)	9.06 (SD = 5.16)	<0.0001
Cycle length (days)	31.29 (SD = 9.54)	30.50 (SD = 8.83)	33.06 (SD = 10.92)	0.2480
VEGF1 (pg/mL)	59.24 (SD = 60.06)	54.47 (SD = 57.94)	72.40 (SD = 64.98)	0.2315

## Data Availability

All data described in the main text.

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
