# Peer review of "How to Improve Non-Invasive Diagnosis of Endometriosis with Advanced Statistical Methods"

_medicina, 2023, doi:10.3390/medicina59030499_

Round 1
Reviewer 1 Report
1. This is a reanalysis of data from 2010 used in a study published in 2014 that concluded that there were “higher plasma concentrations of CA-125, as well as higher concentrations of both CA-125 and VEGF in the peritoneal fluid,” but no “combinations of given markers had a sensitivity >60%.” The current use of Lasso analysis identified the combination of painful menses, BMI levels, and CA125 as predictive of endometriosis with sensitivity and specificity of 0.88 and 0.80, respectively, in a retrospective study of a population with a 70% prevalence of endometriosis.
2. Line 22 and 136. Were the 7 of 19 features based on the results of LASSO chosen prospectively or after analysis of the results? In either situation, the use of 3 of 19 factors and one specific type of analysis (LASSO regression) in a retrospective study appears to be a form of data dredging (https://en.wikipedia.org/wiki/Data_dredging) or HARKing (https://en.wikipedia.org/wiki/HARKing).
3. A statistician needs to be added as an author.
4. The results need to be prospectively confirmed before publication.
5. The results need to be prospectively confirmed in a low-risk population before this is generalizable or useful for primary or early endometriosis.
6. Line 88. Clarify or delete “r” in “was published in 2014 r.”
7. Lines 89-93. “Informed consent for using anonymized data was given by the patients also for further analyses (RNN/11/09/KE).” “Anonymized data” was not a requirement for IRB consideration in the US until 2018. If the informed consent was in 2010, do you still have copies of the consents? Is there a more recent consent for this submission? If you mean “informed consent for using anonymized data was given by the patients,” please send a copy of the consent for review. Do you mean “The use of anonymized data from a 2010 study was approved by the bioethics committee.”?
8. Line 94. Clarify that the laparoscopies were performed in 2010.
9. Line 108-109 Did decision trees, naive Baye’s, or K-means method show significant or non-significant results?
10. Line 109. What is meant by “efficient” in “among them LASSO was the most efficient method.”?
11. Line 126. “76.81% of the study group had painful periods (VAS ≥4)” What was the indication for surgery in the other 23% of patients?
12. Line 126. How long had the patients had painful periods?
13. Lines 194-196 “Our algorithm reached satisfactory sensitivity with only three easy features: painful menses, BMI level and CA125 concentration” is correct in a retrospective study in 18- to 47-year-olds population with a mean age of 31.99 and a 70% prevalence of endometriosis of whom 76.81% had painful periods. This needs confirmation in a prospective study.
14. Line 206-207. What do you mean by a “non-clinical settings where an availability of invasive diagnostic methods is obstructed”?
15. The data and analysis may not be generalizable to infertility, primary care, or adolescent populations. The limitations need to be clarified. Add a section on limitations and discuss the general concepts of selection bias.
16. A discussion of the many studies in which CA125 has not been useful is needed.
17. What was the length of time from the onset of symptoms?
18. What previous treatment had been used?
19. How many patients had stages I, II, III, and IV endometriosis?
20. How many patients had focal tenderness?
21. How many patients had deep infiltrating endometriosis?
22. How many patients had bowel endometriosis?
23. How many patients had distal endometriosis?
24. In your discussion of the limitations, the nature of studies done in tertiary practices needs to be clear.
25. Was Redwine’s near-contact laparoscopy used? Redwine DB. Age-related evolution in color appearance of endometriosis. Fertil Steril. 1987 Dec;48(6):1062-3. PMID: 3678506.
26. Failure to consider bias can lead to an unrecognized lack of generalizability. Stacey Missmer and her associates have a series of studies on biases. Discuss the concepts found in her papers. Those concepts include
- The heterogeneity of inclusion and diagnostic criteria and selection bias [Ghasi] affects not only the variability of frequency and distribution but also the reliability of testing. Selection bias is also influenced by referral reason, acceptance of surgery, access to imaging, access to surgical expertise, hysterectomy status, coexisting conditions, and incidental endometriosis. [Shafrir, Horne]
- Women undergoing laparoscopy for pain differ from those undergoing laparoscopy for infertility. [Shafrir]
- Selection bias is particularly important in adolescents. Only those with the most severe symptoms undergo surgery. [Shafrir]
- Horne AW, Missmer SA. Pathophysiology, diagnosis, and management of endometriosis. BMJ. 2022 Nov 14;379:e070750. doi: 10.1136/bmj-2022-070750. PMID: 36375827.
- Ghiasi M, Kulkarni MT, Missmer SA. Is endometriosis more common and more severe than it was 30 years ago? J Minim Invasive Gynecol. 2020 Feb;27(2):452-461. doi: 10.1016/j.jmig.2019.11.018. Epub 2019 Dec 6. PMID: 31816389.
- Missmer SA. Why so null? Methodologic necessities to advance endometriosis discovery. Paediatr Perinat Epidemiol. 2019 Jan;33(1):26-27. doi: 10.1111/ppe.12540. PMID: 30698886.
- Shafrir AL, Farland LV, Shah DK, Harris HR, Kvaskoff M, Zondervan K, Missmer SA. Risk for and consequences of endometriosis: A critical epidemiologic review. Best Pract Res Clin Obstet Gynaecol. 2018 Jul 3. pii: S1521-6934(18)30109-3. doi: 10.1016/j.bpobgyn.2018.06.001 PMID: 30017581
Author Response
Dear Reviewer,
Please see the attachment - you will find the responses as well as an example of informed consent attached

Reviewer 2 Report
An interesting research, however i would recommend to be reviewed by a statistician before publication.
two tables are named table 1 , the first one :
Table 1. The number of samples randomly assigned to training (70%) and testing (30%) group. need to be explained further
Author Response
Dear Reviewer 2:
Thank you very much for your valuable comments. One of us is also a statistician, we corrected the affiliation.
We corrected also the names of the tables and explained the first of them.
Round 2
Reviewer 1 Report
1. The authors have retrospective reanalyzed data in a population with 71% prevalence of endometriosis with exclusion of inflammatory diseases, cancer, myomata, or other sources of positive CA-125. The use of LASSO analysis was retrospectively chosen after other statistic methods showed no significant results. The current use of Lasso analysis identified the combination of painful menses, BMI levels, and CA125 as predictive of endometriosis with sensitivity and specificity of 0.88 and 0.80, respectively, in a retrospective study of a population with a 70% prevalence of endometriosis and other cause of CA-125 elevation excluded.
2. There are significant changes in the content in this revision and those will be addressed.
3. Please check your document with a spelling and grammar checker. There are spelling errors including algorhithm, delibarations, reviewes, metaanalyses, intelilgence, Sociey, Medicaine, continous, Basinged, an for and, hat for had, bare for bear, Altough, etc.
4. Bioptate is an uncommon variation of biopsy.
5. Chemiotherapy is an uncommon variation of chemotherapy.
6. The grammar could be improved by an editing service.
7. The major limitations of this study include the population with 71% prevalence of endometriosis; exclusion of inflammatory diseases, cancer, myomata, or other sources of positive CA-125; the retrospective nature of the data; and the retrospective choice of a statistical method.
8. Lines 20-21: “Among used statistical methods, LASSO regression - new important statistical tool eliminating the least useful features – was shown to have the best results.” Consider “Among used statistical methods, LASSO regression - new important statistical tool eliminating the least useful features – was the only method to have significant results.”
9. Lines 31-32: “Implementing such an algorithm…” Consider “If confirmed in a prospective study of a general population, implementing such an algorithm…” or if you want to discuss high-risk populations, “If confirmed in a prospective study, implementing such an algorithm in populations with a high risk of endometriosis will allow to cover patients suspected of endometriosis with proper treatment.”
10. Lines 45-46: “Peritoneal endometriosis is the form of subtle changes, usually no larger than 5 mm in the diameter.” Peritoneal endometriosis includes highly visible dark scarred and well as more difficult to recognize nonpigmented, subtle forms.
11. Line 78-79: Reference 19 (Nisenblat et al. 2016 Imaging) is on imaging and I find no discussion of CA-125. (https://www.cochranelibrary.com/cdsr/doi/10.1002/14651858.CD009591.pub2/full. Did you mean to use your ref 31 (Nisenblat et al. 2016 Blood Biomarkers). In that reference the sensitivity and specificity for CA-125 were unimpressive for several cut-offs > 16.0 to 17.6 U/ml: 0.56 and 0.91; > 20.0 U/ml: 0.67 and 0.69; > 25.0 to 26.0 U/ml: 0.73 and 0.70; > 30.0 to 33.0 U/ml: 0.62 and 0.76; and > 35.0 to 36.0 U/ml: 0.40 and 0.91. They did not confirm the use of 15 U/ml.
12. Line 78-79: Reference 20 Kimber-Trojnar et al. includes “Its elevated concentration is observed in patients with cancer of the breast, endometrium and lung, as well as in gastrointestinal and inflammatory conditions. An increased level of CA-125 is the most reliable marker for identification of epithelial ovarian cancer. Its suitability is also tested in endometriosis, an inflammatory disease in which CA-125 is secreted into the circulation by the endometrial and mesothelial cells [24,25,26].” They use a cut-off of 35 U/mL and suggest that CA-125 should be tested in two phases of the cycle. They did not confirm the use of 15 U/ml.
13. Line 156: differentiate appears better than differ
14. Line 275: “This needs to be confirmed in other populations.” Consider “This needs to be confirmed in prospective studies and other populations.”
15. Line 278: “exclusion of the disease without any doubts during surgical procedure” is not a clinical or surgical possibility. We miss endometriosis clinically and surgically for many reasons. Russell first published surgically unrecognized in the ovary in 1899. Badescu et al. (2016) noted that 100% of patients had laparoscopically unrecognized bowel endometriosis, In McGuinness et al. (2020), 23% had laparoscopically unrecognized tubal endometriosis. Koninckx et al. (1996) found that 3 of 21 women with painful nodularity during menstruation had laparoscopically unrecognized retroperitoneal endometriosis. There are many other examples.
Badescu A, Roman H, Aziz M, Puscasiu L, Molnar C, Huet E, Sabourin JC, Stolnicu S. Mapping of bowel occult microscopic endometriosis implants surrounding deep endometriosis nodules infiltrating the bowel. Fertil Steril 2016;105: 430-434 e426.
Koninckx PR, Meuleman C, Oosterlynck D, Cornillie FJ. Diagnosis of deep endometriosis by clinical examination during menstruation and plasma CA-125 concentrations. Fertil Steril 1996, 65:280-287
Longo LD. Classic pages in obstetrics and gynecology. Aberrant portions of the müllerian duct found in an ovary: William Wood Russell Johns Hopkins Hospital Bulletin, vol. 10, pp. 8--10, 1899. Am J Obstet Gynecol. 1979 May 15;134(2):225-6. PMID: 377966.
McGuinness B, Nezhat F, Ursillo L, Akerman M, Vintzileos W, White M. Fallopian tube endometriosis in women undergoing operative video laparoscopy and its clinical implications. Fertil Steril. 2020, 114(5):1040-1048. doi: 10.1016/j.fertnstert.2020.05.026. PMID: 32826047.
Russell WW. Aberrant portions of the Müllerian duct found in an ovary: Johns Hopkins Hospital Bulletin, 1899, 10(Nos 94-95-96)(Jan-Feb-Mar)):8-10, plates I-III.
16. Line 280: "adolescents’ population" is better as "adolescent populations"
17. Line 280-281: The algorithm can not be extended on adolescents’ population or women after hormonal treatment… needs the addition of inflammatory diseases; cancer; large myomata; uterine abnormalities; deep endometriosis or endometriomas on imaging. Those were also exclusion criteria.
18. Line 285-287: What is the scientific intent of “Although it could cause selection bias, one should bare in mind that the proposition of an algorithm to diagnose endometriosis is rather to help to select affected women to schedule them to the first line treatment without any delay.”
19. Line 298-301: Blocking the pituitary-ovarian axis for symptoms dates to the 1960s. It is not a recent trend.
20. Line 301-302: Reduction in symptoms could act as a proof of the presence of endometriosis [41]. I was unable to find this in Becker et al (2022). It may be referring to a similar conclusion in the ACOG Practice Bulletin No. 11 (1999) based on Ling et al. (1999). Ling et al. had a population with an 82% prevalence of endometriosis. 82% with endometriosis had pain relief. 73% with no endometriosis had pain relief. That was not significant. Their study selectin criteria was as successful at predicting endometriosis as GnRH suppression. ACOG Committee Opinion. Number 310 (2005 revered the conclusion and stated that “response to empiric therapy does not confirm the diagnosis of endometriosis.”
ACOG Committee on Practice Bulletins--Gynecology. ACOG Practice Bulletin No. 11: Medical management of endometriosis. Obstet Gynecol. 1999 Dec;94(6):1-14. Retraction in: Obstet Gynecol. 2010 Jul;116(1):211. PMID: 21077410.
American College of Obstetricians and Gynecologists. ACOG Committee Opinion. Number 310, April 2005. Endometriosis in adolescents. Obstet Gynecol. 2005 Apr;105(4):921-7. doi: 10.1097/00006250-200504000-00058. PMID: 15802438.
Ling FW for the Pelvic Pain Study Group. Randomized controlled trial of depot leuprolide in patients with chronic pelvic pain and clinically suspected endometriosis. Obstet Gynecol. 1999 Jan;93(1):51-8. doi: 10.1016/s0029-7844(98)00341-x. PMID: 9916956.
21. Line 320: “to modify risk of endometriosis” implies a therapeutic effect. Consider “to determine the risk of endometriosis.
22. Line 326-326: “the non-invasive algorithm will be always the cheapest and most easily available option.” What does that mean? Does it include treatment?
23. Line 333-336: “Hence since several years patients with the endometriosis symptoms should be scheduled to diagnostic laparoscopy only if hormonal treatment does not alleviate symptoms [41], such an easy algorithm can facilitate the treatment decisions.” I do not understand the sentence. What is the algorithm for negative and positive results if hormonal treatment does not alleviate symptoms?
Author Response
Dear Reviewer,
pleas find attached responses in a Word file,
kind regards,
Maria Szubert

Round 3
Reviewer 1 Report
None